# Impact of Cloud Vertical Structure Perturbations on the Retrieval of Cloud Optical Thickness and Effective Radius from FY4A/AGRI

Jing Sun<sup>1,2</sup>, Yunying Li<sup>1,3</sup>, Hao Hu<sup>4</sup>, Qian Li<sup>1,3</sup>, Chengzhi Ye<sup>5</sup>, Yi-ning Shi<sup>4</sup>, Zitong Chen<sup>1</sup>

- 1 College of Meteorology and Oceanography, National University of Defense Technology, Changsha 410073, China
- 2 China Meteorological Administration Basin Heavy Rainfall Key Laboratory/Hubei Key Laboratory for Heavy Rain Monitoring and Warning Research, Institute of Heavy Rain, China Meteorological Administration, Wuhan 430205, China
- 3 Key Laboratory of High Impact Weather(special), China Meteorological Administration, Changsha 410073, China
- 4 State Key Laboratory of Severe Weather Meteorological Science and Technology, CMA Earth System Modeling and Prediction Centre, Beijing 100081, China
- 5 Institute of Meteorological Sciences of Hunan Province, Hunan Meteorological Bureau, Changsha 410118, China

Correspondence: Yunying Li (ghlyy@mail.iap.ac.cn)

Abstract. The vertical structure of clouds plays a critical role in atmospheric radiative transfer and is a major source of uncertainty in satellite-based retrievals of cloud optical thickness (COT) and cloud effective radius (CER). Most operational algorithms assume a single homogeneous layer, but the biases introduced by this simplification under realistic multilayer conditions remain poorly quantified. This study systematically investigates how perturbations in cloud vertical structure affect COT and CER retrievals using FY4A/AGRI (Advanced Geostationary Radiation Imager) and simulations from Advanced Radiative Transfer Modeling System (ARMS) over central and eastern China during June–August 2018. We designed ten sensitivity experiments by varying water and ice content across single-, double-, and triple-layer cloud configurations to quantify the impact of structural differences on channel reflectance and the COT-CER relationship. The results indicate that upper-level ice clouds significantly mask reflectance from lower water clouds, reducing total reflectance by approximately 50% and leading to systematic retrieval biases: single-layer algorithms underestimate COT at small CER (

200

185

Figure 1. Framework of the COT and CER retrieval algorithm for FY4A AGRI

Table 1. Cloud detection thresholds for FY4A AGRI.

| Channel                                             | Physical significance                    | Threshold |
|-----------------------------------------------------|------------------------------------------|-----------|
| $\frac{T_{0.65} - R_{0.825}}{R_{0.65} + R_{0.825}}$ | Normalized Difference Cloud Index (NDCI) | >0.12     |
| $R_{0.65}$                                          | Reflectance                              | >0.3      |
| $\frac{T_{0.65} - R_{1.61}}{R_{1.61} + R_{0.65}}$   | Normalized Difference Snow Index (NDSI)  | >0.26     |
| $R_{0.47}$                                          | Reflectance                              | >0.15     |

Table 2. Input parameters and grid points of the variables used to build the LUT version of the FY4A AGRI.

| Variables                     | Range                                                              | Number   | Unit   |
|-------------------------------|--------------------------------------------------------------------|----------|--------|
|                               |                                                                    | of grids |        |
| Sun zenith                    | 0,5,10,15,20,25,30,35,40,45,50,55,60,65,70,75,80                   | 15       | degree |
| Satellite zenith              | 0,5,10,15,20,25,30,35,40,45,50,55,60,65,70,75,80                   | 15       | -      |
| Relative Azimuth              | 0,10,20,30,40,50,60,70,80,90,100,110,120,130,140,150,1             | 19       | -      |
|                               | 60,170,180                                                         |          |        |
| Water cloud optical thickness | 0.25, 0.32, 0.4, 0.5, 0.6, 0.8, 1.0, 1.26, 1.58, 1.99, 2.51, 3.16, | 27       |        |
|                               | 3.98,5.01,6.3,7.94,10.0,12.59,15.85,19.95,25.12,31.62,             |          |        |
|                               | 39.81,50.12,63.1,79.4,100.0,125.9,158.5                            |          |        |
| Water cloud effective radius  | 2,4,5,8,11,14,17,20,26,30,36,42,50,60                              | 12       | μm     |
| Ice cloud optical thickness   | 0.25, 0.32, 0.4, 0.5, 0.6, 0.8, 1.0, 1.26, 1.58, 1.99, 2.51, 3.16, | 27       |        |
|                               | 3.98,5.01,6.3,7.94,10.0,12.59,15.85,19.95,25.12,31.62,             |          |        |
|                               | 39.81,50.12,63.1,79.4,100.0,125.9,158.5                            |          |        |
| Ice cloud effective radius    | 5,10,15,25,30,35,40,50,60,70,80,90,110,130                         | 12       | μm     |

**Figure 2.** Bispectral reflectance LUTs of cloud reflectance at 0.65 um and 1.61 um for water(a) and ice(b) phases when the solar zenith angle is 40°, the sensor viewing zenith angle is 40°, and the relative azimuth angle is 20°, and the underlying surface is land. The dashed line represents the CER, and the solid line represents the COT.

Figure 3. Schematic of the DORF model for COT and CER prediction based on FY-4A/AGRI observations.

## 3 COT and CER Retrievals and Comparisons

To evaluate the accuracy of FY4A/AGRI cloud property retrievals, we compared them with the MODIS Collection 6.1 MOD06 daytime cloud products from June to August 2018. For spatial matching, the 1 km MODIS pixels were averaged within each 4 km AGRI footprint to ensure consistent resolution. For temporal matching, the MODIS overpass was paired with the closest AGRI full-disk scan (15 min interval), with a maximum offset of ±7.5 min. The FY-4A/AGRI-derived CER exhibit strong consistency with the MODIS MOD06 results, with a coefficient of determination (R²) of 0.91, a mean absolute error (MAE) of 2.0 µm, and a root-mean-square error (RMSE) of 3.36 µm in Fig. 4. Similarly, the FY4A/AGRI-derived COT exhibits good consistency with MODIS MOD06, with R², MAE, and RMSE values of 0.87, 3.9, and 8.03, respectively.

Fig. 5 display the probability density distributions (PDFs)of CER and COT generated from the same 4 km resampled data to ensure consistency with Fig. 4. These PDFs provide a basis for comparing the retrievals between FY4A/AGRI and MODIS. The results indicate that FY4A/AGRI underestimates CER relative to MODIS for CER

These differences mainly result from: (1) spatial resolution differences (MODIS 1 km vs. AGRI 4 km); (2) cloud horizontal inhomogeneity within AGRI pixels; and (3) visible channel degradation and SWIR fluctuations (Sun et al., 2025). Additionally, in the region 106–107°E, 32–35°N (corresponding to the Dabie and Wuling Mountains), FY4A/AGRI and MODIS retrievals show noticeable differences. This discrepancy is likely due to the influence of high-elevation terrain on satellite observations and the fact that the retrieval algorithm was primarily tuned for lowland surface types, without explicitly considering mountainous surface characteristics. Despite this local variation, the overall COT and CER patterns remain consistent within the common observed area. This case thus provides a robust reference for the perturbation experiments in Section 4.

Figure 4. Scattering plot (a and b) of COT and CER from MODIS C6 (MOD06) and FY4A/AGRI data in the selected area.

Figure 5. Probability density function (PDF) of the FY4A/AGRI retrieval results and the MODIS cloud products in the region. The red and black solid line shows the FY4A/AGRI results and the MODIS results, respectively.

**Figure 6.** Comparison of retrieved optical parameters using the FY-4 AGRI with MODIS cloud products (Cloud\_Optical\_Thickness\_16 and Cloud\_Effective\_Radius\_16). The observation time of the FY-4 AGRI is 04:00 UTC on 2 August 2018 and the MODIS observation time is 03:35 UTC. (a) False-color image (red, 0.65 μm; green, 1.61 μm; blue, 10.7 μm reversed) where thick ice clouds are orange colored, and low clouds are white colored. (b, d, f) cloud-top phase, COT(unitless), CER(unit: μm) from Collection-6.1 MOD06 at 0335 UTC August 2, 2018. (c,e) COT(unitless), CER(unit: μm) derived FY4A/AGRI at 0400 UTC August 2, 2018.

# 4 Cloud Microphysical and Radiation Response to Cloud Vertical Structure

Cloud vertical structure includes the number of cloud layers, cloud top height, cloud base height, cloud thickness, cloud fraction, and the vertical distribution of cloud microphysical properties. It reflects the thermodynamic, dynamic, and microphysical processes within the cloud system and plays an

important role in weather and climate (Xu et al., 2023). To quantitatively investigate the effects of cloud layer number and cloud phase on cloud reflectance, COT and CER, this study builds upon the case study retrieval results from Section 3.2. Cloud water profiles are categorized into single-layer, double-layer, and triple-layer clouds, and by adjusting the cloud liquid/ice water content in each layer, the sensitivity of cloud reflectance, COT and CER under different vertical cloud structure conditions is systematically analyzed.

## 4.1 Cloud water/ice content Profiles Classification

280

285

To investigate the impact of cloud vertical structure on COT and CER, 221 vertical profiles of cloud liquid and ice water content were extracted from ERA5 at 04:00 UTC on 2 August 2018 over the region 110~114°E and 30~33°N (Section 3.2). Based on empirical rules, each profile was classified into different structural types. The number of peaks in liquid/ice water content was used to determine the number of cloud layers, while cloud phase was inferred by pressure level: ice clouds above 450 hPa, and liquid clouds below 700 hPa and between 700–450 hPa.

Specifically, single-layer clouds, characterized by one peak, were divided into three types: high-level ice clouds, mid-level water clouds, and low-level water clouds. Two-layer cloud structures, identified by two peaks, were further categorized into five subtypes based on the cloud phase (ice or water) and the relative liquid or ice water content in the upper, middle, and lower layers. Three-layer profiles, indicated by three peaks, generally represent a typical ice—water—water cloud configuration (Fig. 7).

Statistical results (Table 3) show that single-layer clouds account for 48% of the profiles, two-layer clouds 46%, and three-layer clouds 6%. These proportions are consistent with the findings of Xu et al. (2023), who reported that single-layer clouds dominate (55.4%) in radiosonde observations, with two-layer systems being the most frequent among multilayer clouds. This agreement supports the validity and physical relevance of the classification method used.

**Figure 7.** Vertical structure types of cloud water and ice content profiles from ERA5. The "large cloud" profiles correspond to the original profiles with a fivefold increase in cloud water or ice content, while the "small cloud" profiles represent the unmodified original profiles. The distinction between large and small clouds is not explicitly marked in the figure but can be inferred from the differences in water content.

**Table 3** Proportions of single-layer, two-layer, and three-layer Cloud Profiles and Corresponding Mode Classifications

| Cloud     | Height | Cloud  | Exp  | Ratio |
|-----------|--------|--------|------|-------|
| Layers    |        | type   | No.  |       |
| Single    | Н      | Ice    | 3    |       |
| Layers    | M      | Water  | 2    | 48%   |
| Cloud     | L      | Water  | 1    |       |
|           | H+M    | Ice +  | 8    |       |
| Two-layer |        | Water  |      | 46%   |
| Clouds    | H+L    | Ice +  | 6,7  |       |
|           |        | Water  |      |       |
|           | L+M    | Ice +  | 4,5  |       |
|           |        | Water  |      |       |
| Three-    | H+M+L  | Ice +  | 9,10 | 6%    |
| layer     |        | Water  |      |       |
| Clouds    |        | +Water |      |       |

## 4.2 Sensitivity of Reflectance to Cloud Vertical Structure

Based on the classification, the sensitivity experiments are designed to investigate how perturbations in cloud liquid water content (CWC) and ice water content (IWC) at different vertical levels (low, mid, and high) influence the reflectance responses of FY-4A/AGRI Channels 2 and 5. The vertical positions of the cloud layers are prescribed and fixed, so the variations in reflectance shown in Figs. 8–10 result from systematic changes in CWC and IWC at these levels, rather than from changes in cloud height alone. To ensure representativeness of each cloud type, the CWC and IWC vertical profiles used in the experiments were constructed from the mean profiles of ERA5 for each category.

Fig. 8 shows the sensitivity of cloud reflectance to CER variations for three single-layer cloud types under increased LWC or IWC. For low-level water clouds, channel 2 reflectance rises from 0.6 to 0.7 as CER increases from 2 to 10 μm, while channel 5 first decreases then rise to 0.68 (Fig. 8a). Increasing LWC enhances channel 2 reflectance by about 0.1, with a smaller effect on channel 5. Mid-level water clouds exhibit significant reflectance changes in channel 2 only when CER exceeds 10 μm, stabilizing beyond 25 μm (Fig. 8b). High-level ice clouds show slightly higher reflectance in both channels compared to water clouds (Fig. 8c). Overall, increased LWC in low-level clouds notably boosts reflectance, especially in the visible channel, by approximately 15% more than mid- or high-level clouds. Moreover, the minimal response of channel 5 reflectance for high-level ice clouds when increasing IWC is consistent with Wang et al. (2018), who showed that shortwave infrared (SWIR) channels are primarily weighted toward cloud top. If the top-layer CER is already large, additional IWC exerts little effect on SWIR reflectance.

For double-layer clouds, the reflectance response is determined by the combination of liquid and ice clouds. In the "mid-level water cloud–high-level ice cloud" configuration, channel 5 reflectance remains nearly unchanged under different perturbations, whereas channel 2 exhibits significant variations. For CER < 5 µm, increasing mid-level CWC or high-level IWC yields almost identical results; for CER > 5 µm, increasing high-level IWC produces slightly higher channel 2 reflectance than increasing mid-level CWC (Fig. 9a, d), likely due to multiple scattering in the upper ice cloud enhancing upward radiation while partially diminishing the mid-level water cloud contribution. In the "low-level water cloud–high-level ice cloud" configuration (Fig. 9c and f), the overall reflectance response is similar to that of the mid-level water cloud–high-level ice cloud case, indicating that the upper ice cloud dominates the system's radiative properties. This mechanistically supports the observational findings of Kiran et al. (2015), which reported that despite the presence of liquid water clouds at lower levels, the net radiative

forcing at the top of the atmosphere remains nearly balanced, primarily due to the simultaneous shortwave cooling and longwave heating effects of the upper ice cloud. By contrast, in the "low-level water cloud–mid-level ice cloud" configuration, the reflectance response to increases in CWC/IWC at different levels is markedly different from the previous two cases. Specifically, increasing low-level CWC causes channel 2 reflectance to vary nonlinearly with CER, first decreasing and then increasing (Fig. 9b), whereas increasing mid-level IWC results in a linear decrease of reflectance in both channels 2 and 5, with channel 5 even decreasing by approximately 0.12 when CER > 30 µm (Fig. 9e).

For the three-layer clouds consisting of high-level ice cloud over mid-level water cloud over low-level water cloud, increasing either low- or mid-level CWC enhances Channel 2 reflectance with increasing CER, with the increase being more pronounced for mid-level CWC (Fig. 10a-b). The response of Channel 5 reflectance exhibits a similar trend but with a smaller magnitude, indicating that the mid-level water cloud contributes more significantly to reflectance in both channels. When high-level IWC is increased, Channel 2 reflectance decreases approximately linearly with CER, while Channel 5 also declines (Fig. 10c). For CER >14 μm, Channel 5 reflectance drops from ~0.4 to 0.25, highlighting the strong radiative shielding effect of the high-level ice cloud on underlying water clouds. These observations align with Li et al. (2011), who found that multilayer clouds have weaker shortwave reflectance than single-layer clouds due to their higher cloud tops allowing shortwave radiation to partially transmit to lower clouds or the surface.

**Figure 8.** Variation of cloud reflectance at Channel 2 (solid lines) and Channel 5 (dashed lines) with cloud effective radius (CER) under different LWC or IWC condition. Black lines correspond to reflectance simulated using the original LWC/IWC profiles from ERA5 reanalysis, while red lines represent reflectance simulated with LWC/IWC increased by a factor of 5. Three single-layer cloud types are shown: low-level water clouds (a, Exp 1), mid-level water clouds (b, Exp 2), and high-level ice clouds (c, Exp 3). The simulations assume a solar zenith angle of 40°, sensor viewing zenith

angle of 40°, and relative azimuth angle of 20°.

**Figure. 9.** Reflectance—CER relationships for six double-layer cloud sensitivity experiments: (a, d) mid-level water + high-level ice cloud, with (a) increased mid-level CWC (Exp 8) and (d) increased high-level IWC(Exp 8); (b, e) low-level water + mid-level ice cloud, with (b) increased low-level CWC(Exp 4) and © increased mid-level IWC(Exp 5); (c, f) low-level water + high-level ice cloud, with (c) increased low-level CWC(Exp 6) and (f) increased high-level IWC(Exp 7). Each panel illustrates the effect of adjusting cloud water content (CWC) or ice water content (IWC) on the reflectance—CER relationship for the corresponding cloud vertical structure.

Figure 10. Similar to Fig. 9, but for multi-layer clouds (a) Exp 9 and (b) Exp 10.

## 4.3 Sensitivity of COT-CER Relationship to Cloud Vertical Structure

The operational COT and CER retrieval algorithm developed in Section 3 is based on single-layer cloud assumptions. In reality, cloud systems are frequently multi-layered, which can introduce significant uncertainties in retrievals. To quantify these effects, we conducted idealized perturbation experiments using representative single-, double-, and three-layer cloud configurations (Fig. 11), perturbing CWC or

360

IWC at specific vertical levels and comparing the resulting optical thickness ( $\Delta$ COT) with unperturbed cases.

For single-layer clouds,  $\Delta$ COT increases nonlinearly with CER, showing a rapid rise when CER is below 15 μm before approaching saturation. Notably, when CER < 10 μm, increasing LWC in mid-level water clouds leads to a maximum  $\Delta$ COT of 52, which is approximately 1.6 times greater than that resulting from the same perturbation in low-level clouds (Fig. 11b). For two-layer structures, three typical configurations were analyzed: (1) mid-level ice cloud over low-level water cloud (Fig. 11c), (2) highlevel ice cloud over low-level water cloud (Fig. 11d), and (3) high-level ice cloud over mid-level water cloud (Fig. 11e). On average, the COT increase due to low- and mid-level water cloud variations in single-layer clouds exceeds that in double-layer clouds by about 24%, primarily due to the masking effect of upper-level ice clouds in double-layer structures. Low-level water beneath mid-level ice showed negative ΔCOT at small CER, whereas high-level ice over low-level water enhanced ΔCOT. The highlevel ice over mid-level water scenario exhibited a non-monotonic ΔCOT pattern for small CER, likely due to complex radiative interactions between layers. For three-layer clouds, increases in mid- or lowerlevel CWC significantly enhance ΔCOT, although peak values are slightly lower than in single-layer clouds (Fig. 11e). The lower water layer dominates the response, while overlying ice partially masks the effect. These results agree with Wang et al. (2021), who showed that subgrid-scale cloud structure and overlapping condensate significantly modulate radiative effects.

We further examined the impact of multi-layer cloud vertical structures by comparing COT from idealized double- and triple-layer simulations with single-layer retrievals. Fig. 12 shows the difference between COT retrieved under the single-layer assumption and that simulated with multi-layer clouds ( $\Delta$ COT\_retrieval, hereafter abbreviated as  $\Delta$ COT\_R) as a function of CER. Overall, when CER < 10  $\mu$ m,  $\Delta$ COT\_R changes from negative to positive, indicating that the single-layer assumption systematically underestimates the true COT under small droplet conditions. As CER increases beyond 14  $\mu$ m,  $\Delta$ COT\_R gradually becomes positive, with single-layer retrievals exceeding two-layer simulations by approximately 20 units on average. This primarily results from the single-layer assumption's inability to capture the shielding effect of overlying ice clouds on underlying water clouds, as well as the differential contribution of particles at different vertical levels to reflectance in the visible and shortwave infrared channels.

For the "mid-level water cloud and high-level ice cloud" structure, ΔCOT R remains negative for

CER < 22  $\mu$ m before turning positive (Fig. 12a). For the "low-level water cloud and high-level ice cloud" structure, positive  $\Delta$ COT\_R appear only when CER > 45  $\mu$ m (Fig. 12b). Increasing the IWC of the high-level ice cloud maintains negative  $\Delta$ COT\_R at small CER, with the positive transition also at CER > 45  $\mu$ m. In the "low-level water and mid-level ice" scenario,  $\Delta$ COT\_R is near zero for CER 

**Figure 11.** Changes in cloud optical thickness ( $\Delta$ COT) as a function of CER for six selected vertical cloud structure types (Exp 1, 2, 7, 8, and 10). Each panel shows the  $\Delta$ COT resulting from adding LWC or IWC to a specific vertical layer relative to a reference state without that layer. Blue lines represent the contribution of added LWC, while red lines represent added IWC. Model numbers are

indicated in each panel. when the solar zenith angle is  $40^{\circ}$ , the sensor viewing zenith angle is  $40^{\circ}$ , and the relative azimuth angle is  $20^{\circ}$ .

Figure 12. Differences in cloud optical thickness (COT) between multilayer cloud vertical structures and the single-layer assumption as a function of CER. Blue: difference between COT retrieved under the single-layer assumption and COT simulated for double-layer clouds; Red and pink: difference after adding CWC to the mid-level water cloud; Green: difference after adding IWC to the high-level ice cloud.

## 425 5 Summary and Conclusions

430

435

Based on FY4A/AGRI geostationary satellite observations over eastern and central China during June–August 2018, this study focuses on the influence of cloud vertical structure on the relationship between cloud optical thickness (COT) and cloud effective radius (CER). A bispectral retrieval algorithm for COT and CER was developed using the Advanced Radiative Transfer Modeling System (ARMS) and validated against MODIS cloud products. To systematically examine the impact of vertical cloud heterogeneity, ten idealized single-, double-, and three-layer cloud structures were constructed, allowing evaluation of how perturbations in cloud water content (CWC) and ice water content (IWC) at different altitudes affect both reflectance and the resulting COT–CER relationship.

The COT and CER retrieved from FY4A/AGRI show good agreement with MODIS, confirming its reliability for analyzing the sensitivity of COT and CER to cloud vertical structure. Figure 13 presents a conceptual schematic of the radiative effects of cloud layering. Cloud reflectance in the visible and

shortwave-infrared channels is influenced by CWC and IWC at different altitudes, as well as particle size. For single-layer liquid clouds, increasing low-level LWC strongly enhances reflectance, with visible channel increases about 15% greater than those for mid-level liquid or high-level ice clouds. In two-layer clouds, the "mid-level water—high-level ice" and "low-level water—high-level ice" configurations are dominated by the upper ice cloud, while in the "low-level water—mid-level ice" configuration, increasing low-level CWC first reduces and then increases reflectance, whereas increasing mid-level IWC causes a linear decrease. For three-layer clouds, mid- and low-level water clouds contribute most to the visible channel, while high-level IWC strongly reduces reflectance through radiative shielding.

Through idealized perturbation experiments on representative single-, double-, and triple-layer cloud structures, we quantified the impact of cloud vertical structure on the COT–CER relationship. Results show that for single-layer clouds,  $\Delta COT$  increases nonlinearly with CER, reaching a maximum at small particle sizes (CER < 15  $\mu m$ ). When CER < 10  $\mu m$ , an increase in mid-level water cloud LWC produces a  $\Delta COT$  of up to 52, approximately 1.6 times that of an equivalent perturbation in low-level clouds. For multi-layer clouds, overlying ice layers partially mask the radiative effects of lower water clouds, causing  $\Delta COT$  to be negative or non-monotonic at small CER. Further comparison between multi-layer simulations and single-layer retrievals shows that single-layer retrievals systematically underestimate COT at small CER, while at larger CER, the single-layer results exceed double-layer simulations by about 20 units on average. For the low-level water–mid-level ice cloud structure, the difference between retrieval and simulated results is close to 0 when CER < 5  $\mu m$ , gradually increases with CER, and plateaus for CER > 30  $\mu m$ , reflecting the sustained shielding effect of the upper ice layer on the lower water cloud. In triple-layer clouds, increasing mid-level CWC causes single-layer retrievals to remain below simulation values for CER 

Figure 13. Conceptual diagram illustrating the radiative characteristics and retrieval implications of COT and CER under different vertical cloud structures.

Code availability. The code used in this study are available from the corresponding author upon reasonable request(ghlyy@mail.iap.ac.cn).

Data availability. The FY4A/AGRI data used for the main COT and CER retrieval in this study are 475 released from the Fengyun Meteorological Satellite Remote Sensing Data Service Platform (https://satellite.nsmc.org.cn/DataPortal/cn/data/order.html). The MODIS cloud product used for the DORF model building and for validating the spatial distributions of retrieved COT and CER from FY4A/AGRI(https://ladsweb.modaps.eosdis.nasa.gov/search/order/1/MOD06 L2--61,MYD06 L2--61). The ERA5 reanalysis data provided by the European Centre for Medium-Range Weather Forecasts 480 (ECMWF) are used to supply atmospheric temperature, humidity, and pressure profiles as input to the ARMS model (Hersbach et al., 2020). Additionally, vertical profiles of cloud liquid water and ice water content from ERA5 are used to construct and classify the idealized cloud vertical structure models used in the sensitivity experiments (https://cds.climate.copernicus.eu/datasets/). The Advanced Radiative Transfer Modeling System (ARMS) model developed in China (Weng et al., 2020), and the package is 485 available from hanyang@cma.gov.cn. The random forest technique is available at https://scikitlearn.org/stable/modules/generated/sklearn.ensemble.

**Author contributions.** JS conceptualized the study, and wrote the original draft. JS and YL designed the algorithms, and revised the manuscript. HH, QL, and CY validated results and supervised the research. YS and ZC performed the simulations.

Competing interests. The authors declare that they have no known competing financial interests or

personal relationships that could have appeared to influence the work reported in this paper.

Acknowledgements. The authors would like to thank the National Satellite Meteorological Center for providing the FY4A/AGRI Level-1 full-disk observation data, the National Aeronautics and Space Administration (NASA) Level-1 and Atmosphere Archive & Distribution System (LAADS) for providing the MODIS cloud products, the European Centre for Medium-Range Weather Forecasts (ECMWF) for the ERA5 reanalysis data. We also acknowledge the Numerical Weather Prediction Center of the China Meteorological Administration (CMA) for the development of the Advanced Radiative Transfer Modeling System (ARMS).

Financial support. This work was jointly supported by the Hunan Provincial Natural Science Foundation of China (grant number 2021JC0009), the National Natural Science Foundation of China (grant number U2242201), and the Hubei Provincial Natural Science Foundation of China (2025AFD423).

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
