# Peer review of "Impact of Cloud Vertical Structure Perturbations on the Retrieval of Cloud Optical Thickness and Effective Radius from FY4A/AGRI"

_EGUsphere, 2025_

## Author Comment (AC2)

**Response to the Reviewer1**

Clouds play an important role in atmospheric radiation (thus Earth energy budget), weather and climate. Among various cloud properties, cloud optical thickness (COT) and effective radius (CER), which can be revealed by reflected solar radiation, have been widely used to infer cloud radiative effects. Thus, COT and CER have become almost standard cloud products of satellite imagers, and, among them, MODIS provided one of the most popular ones. This manuscript by Sun et al. presented numerical study on the influences of cloud vertical inhomogeneity on the retrieval of cloud COT and CER using AGRI observations. The manuscript is overall clearly organized and presented, while there are some major issues that should be considered before it could be considered for publication.

**Response:** We sincerely thank the reviewer for their careful reading of our manuscript and for the constructive comments, which have greatly helped us improve the clarity and quality of the paper. Below, we provide our responses to the specific comments raised. The issues raised by the reviewer have been addressed (in blue color) in the revised manuscript. Kindly find a point-by-point reply to the issues as follows (presented in blue color).

Major Comments:

1. The focus of this study is suggested to be clarified. This study mainly investigated the influences of vertical structures perturbation on the retrieval, while the use of a particular kind of satellite observations, i.e., FY4A/AGRI, for COT and CER is not new at all. The COT and CER algorithm was the classic Nakajima and King (1989) algorithm, and Liu et al. (2023) (referred in the manuscript) presented the operational FY4A AGRI COT and CER retrieval. Thus, the key contributions of this study are the investigation on vertical structure.

**Response:** We sincerely thank the reviewer for this constructive comment. We agree that the use of FY4A/AGRI observations and the Nakajima and King (1990) algorithm itself does not constitute the novelty of this study. In the revised manuscript, we have explicitly emphasized in the Abstract (**Lines 18~35**), Introduction (**Lines 49~60, Lines 70~97**), and Conclusions (**Lines 426~469**) that the primary objective is to systematically quantify the impact of cloud vertical structure perturbations on the

retrievals of cloud optical thickness (COT) and cloud effective radius (CER). Since the retrieval results serve as the basis for the sensitivity experiments, we have streamlined the retrieval sections in the revised manuscript by merging Sections 3.1 and 3.2(**Lines 214~246**).

Moreover, we have included a detailed analysis comparing the COT–CER relationships obtained from multi-layer cloud simulations with those derived under the single-layer assumption, highlighting the systematic biases introduced by neglecting vertical heterogeneity. These revisions clearly underscore the core contribution of the study and directly address the reviewer's concern.

2. Actually, the used of FY4A introduces more "uncertainties". A recently study indicated that the operational AGRI L1 radiance data themselves may be less reliable due to the calibration degradation (https://ieeexplore.ieee.org/document/11071868). Such uncertainties should be considered in the evaluation of the AGRI results.

**Response:** We sincerely thank the reviewer for highlighting potential calibration uncertainties in FY-4A/AGRI data, which are indeed important to consider in COT and CER retrievals. In the revised manuscript (**Section 2.1, Lines 119–131**), we have added a discussion on radiometric calibration, geometric correction, and known uncertainties based on Sun et al. (2025).

Specifically, this study uses FY-4A/AGRI Level-1 (L1) full-disk data. For each channel, digital numbers (DNs) are converted to physical quantities using official calibration information, and interpolated to the study grid if necessary. Procedures follow official methods:

(1) **Calibration:** Linear calibration coefficients are applied as:

$$\text{Top-of-atmosphere (TOA) Reflectance or Radiance} = k \cdot DN + b$$

for the $i_{th}$ channel, the $i_{th}$ row provides (k, b).

In the COT/CER retrieval, we mainly use the TOA reflectance of solar reflective channels.

(2) **Geometric correction and spatial interpolation:** Nominal AGRI geometry is used to map geographic regions to line/column numbers (latlon2linecolumn). Nearestneighbor interpolation is applied to the study grid to avoid excessive smoothing or mixed-pixel effects. These steps are consistent with official procedures

(3) **Quality control and missing value handling:** Fill values, saturated, and anomalous pixels are masked. Visual inspections are conducted over land/sea edges and strong scattering areas (e.g., glint) to ensure data quality.

(4) **Calibration uncertainties:** Recent assessments indicate that FY-4A/AGRI exhibits long-term degradation in the visible channels (0.47 and 0.65 μm), whereas the SWIR channels (1.61 and 2.25 μm) show smaller long-term degradation but larger temporal fluctuations. Using the cloud-target (CT) based recalibration method, the overall calibration accuracy can be controlled within ±3% (Sun et al., 2025).

Considering that the data used in this study are from June–August 2018 (early FY-4A mission, relatively short time span), we adopted the following measures to mitigate the impact of calibration uncertainty on retrievals:

- Calibrate using the official coefficients and lookup tables to ensure consistency with operational procedures (https://satellite.nsmc.org.cn/DataPortal/cn/home/index.html);

- Independently validate AGRI-derived COT and CER against MODIS cloud products (see Section 3.1), which show good agreement;

- Evaluate the magnitude of the impact of cloud vertical structure perturbations on the reflectance in visible and shortwave near-infrared channels using ARMS sensitivity experiments. The results show that the reflectance and retrieval biases induced by vertical structure perturbations are consistent with the findings for the typical two-layer cloud system composed of upper-level ice clouds and lower-level water clouds, which is the most prevalent type, accounting for over 50% of overlapping cloud cases (Sourdeval et al., 2016).

These analyses indicate that FY-4A/AGRI calibration uncertainties do not alter the core conclusions regarding COT/CER biases caused by vertical cloud structure perturbations, and the data are suitable and reliable for this study. We believe that this addition significantly improves the transparency and rigor of our manuscript by

proactively acknowledging and discussing the data limitations. We are grateful to the reviewer for guiding us to strengthen this aspect of our paper.

**References:**

[1] Sun, C., Liu, C., Lu, F., et al. Examination of Long-Term Fengyun-4 AGRI Reflective Solar Bands Calibration Using Cloud Targets. IEEE Transactions on Geoscience and Remote Sensing, 63.DOI:10.1109/TGRS.2025.3585943, 2025.

[2] Sourdeval, O., Laurent, C-L., Baran, A. J., and Gérard, B.: A methodology for simultaneous retrieval of ice and liquid water cloud properties. Part 2: Near-global retrievals and evaluation against A-Train products. Q. J. R. Meteorol. Soc. 142(701): 3063-3081, https://doi.org/10.1002/qj.2405, 2016.

3. Excellent consistency was noticed in Figure 4 between the results of MODIS and AGRI, while the direct comparisons in Figures 5 and 6 shows clear differences. Such significant differences should be carefully checked. Again, considering the uncertainties on AGRI calibration as well as data collocation, the agreement in Figure 4, which is much better than results in Figures 5 and 6, should be exampled.

**Response:** We sincerely appreciate the reviewer's comment. To further clarify this point, we have plotted the PDFs of COT and CER for MODIS (1 km) and FY-4A/AGRI (4 km) in **Fig. S1**. The results indicate that the overall trends of the two datasets are generally consistent, , supporting the robustness of our retrievals.

The apparent discrepancy between Figs. 4–6 arises mainly from the different comparison strategies: Fig. 4 presents a scatterplot between MODIS data resampled to 4 km and AGRI retrievals, highlighting their overall pixel-wise correlation; Fig. 5 compares PDFs on the common 4 km grid, illustrating statistical characteristics; and Fig. 6 shows spatial distributions at the original resolutions, which naturally emphasize differences due to sensor resolution and retrieval algorithms (**Lines 214~220, Lines 221~225, Lines 230~237, in the revised manuscript**).

Previous studies have shown that cross-resolution data matching may introduce a "partial-filling effect." For example, when higher-resolution visible pixels (~2 km) are matched to coarser radar pixels (~5 km), clear-sky areas may be included, leading to

shifts in the PDFs (Chen and Fu, 2017; Chen et al., 2020). Overall, the observed differences are mainly attributable to: (1) spatial resolution differences (MODIS 1 km vs. AGRI 4 km); (2) horizontal inhomogeneity of clouds within AGRI pixels; and (3) visible channel degradation and SWIR fluctuations (Sun et al., 2025). In addition, in the region of 106–107°E and 32–35°N (corresponding to the Dabie and Wuling Mountains), FY-4A/AGRI and MODIS retrievals exhibit noticeable differences (Fig.6), which may be related to the influence of high-elevation terrain on satellite observations. The current retrieval algorithm was primarily tuned for lowland surface types and does not explicitly account for mountainous characteristics. Despite these local discrepancies, the overall distributions of COT and CER remain consistent across the overlapping regions, thereby confirming the robustness and reliability of the retrieval method and providing strong support for the sensitivity experiments in Section 4(**Lines 238~246 in revised manuscript**).

[Figure]

**Fig. S1.** Probability density function (PDF) of the FY4A/AGRI retrieval results and the MODIS cloud products in the region. The red and blue solid line shows the FY4A/AGRI(4km) results and the MODIS(1km) results, respectively.

**References:**

[1] Fu, Y.: Cloud parameters retrieved by the bispectral reflectance algorithm and associated applications, J. Meteorol. Res., 28, 965982, https://doi.org/10.1007/s13351-014-3292-3, 2014.

[2] Ackerman, S. A., Holz, R. E., Frey, R., Eloranta, E. W., Maddux, B. C., and Mcgill, M.: Cloud detection with MODIS. Part II: Validation, J. Atmos. Ocean. Tech., 25, 1073–1086, https://doi.org/10.1175/2007JTECHA1053.1, 2008.

[3] Chen, Y., Chen, G., Cui, C., Zhang, A., Wan, R., Zhou, S., Wang, D., and Fu, Y.: Retrieval of the vertical evolution of the cloud effective radius from the Chinese FY-4(Feng Yun 4) next-generation geostationary satellites. Atmos. Chem. Phys. 20,1131-1145, https://doi.org/10.5194/acp-20-1131-2020, 2020.

4. Some key previous studies on the influences of vertical structure on COT and CER retrievals should be mentioned and discussed. For example, Wang et al., did a systematic study using MODIS observations (https://doi.org/10.1029/2018JD029681).

**Response:** We thank the reviewer for the suggestion. Following your comment, we have added references and discussion of key previous studies on the influence of vertical cloud structure on COT and CER retrievals. Specifically:

**Lines 311~314:** We added the statement: "the minimal response of channel 5 reflectance for high-level ice clouds when increasing IWC is consistent with Wang et al. (2018), who showed that shortwave infrared (SWIR) channels are primarily weighted toward cloud top. If the top-layer CER is already large, additional IWC exerts little effect on SWIR reflectance." This mechanism explains why in our experiments, channel 5 responds minimally to increased IWC, whereas the visible channel (COT-sensitive) still shows noticeable enhancement.

**Lines 321~327:** We added the statement: "In the "low-level water cloud–high-level ice cloud" configuration (Fig. 9c and f), the overall reflectance response is similar to that of the mid-level water cloud–high-level ice cloud case, indicating that the upper ice cloud dominates the system's radiative properties. This mechanistically supports the observational findings of Kiran et al. (2015), which reported that despite the presence of liquid water clouds at lower levels, the net radiative forcing at the top of the atmosphere remains nearly balanced, primarily due to the simultaneous shortwave cooling and longwave heating effects of the upper ice cloud."

**Lines 382~384:** We added the following discussion: "These results agree with Wang et al. (2021), who showed that subgrid-scale cloud structure and overlapping condensate significantly modulate radiative effects." Wang et al. showed that subgrid-scale structural parameters significantly modulate cloud radiative effects, and the overlapping cloud condensate exerts a non-negligible influence on radiative transfer process. Our results can be interpreted as a layer-level verification of this mechanism: the masking effect of overlying ice clouds is analogous to the modulation of radiative transfer, while the strong ΔCOT response of low-level water clouds at small particle sizes reflects the influence of overlapping cloud condensate.

**References:**

[1] Wang, C., Platnick, S., Fauchez, T., et al. An Assessment of the Impacts of Cloud Vertical Heterogeneity on Global Ice Cloud Data Records From Passive Satellite Retrievals. Journal of Geophysical Research: Atmospheres, 124(3):1578-1595. DOI:10.1029/2018JD029681, 2019.

[2] Ravi Kiran, V., Rajeevan, M., Gadhavi, H. et al. Role of vertical structure of cloud microphysical properties on cloud radiative forcing over the Asian monsoon region. Climate Dynamics, 45, 3331–3345, https://doi.org/10.1007/s00382-015-2542-0,2015.

[3] Wang, X., Miao, H., Liu, Y., et al. Dependence of cloud radiation on cloud overlap, horizontal inhomogeneity, and vertical alignment in stratiform and convective regions. Atmospheric Research, 249:105358-105269, 2021.

5. Actually, Teng et al. (2020) did not try to show that ice-over-water system give more consistent results with observations. They developed a much advanced algorithm to retrieval COTs and CERs of ice-over-water clouds, which is almost a new retrieval algorithm. Meanwhile, their algorithm has been improved to infer cloud top heights as well (https://doi.org/10.1016/j.rse.2022.113425). The ideas of Teng et al. (2020 and 2023) are quite different from the presented study, and should be clarified.

**Response:** We thank the reviewer for pointing out the need to clarify the distinction between our study and the work of Teng et al. (2020, 2023). We fully agree that their

studies pursued fundamentally different objectives and employed substantially different methodologies.

We have revised the manuscript (**Lines 74~80 in the revised manuscript**) to explicitly state this distinction: Teng et al.'s work is cited for contextual purposes only, while our study is based on a fundamentally different methodological framework and addresses a different research objective. We appreciate this valuable suggestion, which has helped improve the clarity and precision of our presentation.

6. What's the physical reasons for the differences on reflectance of L and M water clouds (i.e., Exp. 1 and Exp. 2) if the only differences was the cloud location.

**Response:** We thank the reviewer for pointing out the reflectance differences between low-level and mid-level water clouds. In our sensitivity experiments, we systematically examined the effects of perturbing cloud liquid water content (CWC) and ice water content (IWC) at different altitudes (low, mid, and high) through various combinations of water and ice clouds. Thus, the reflectance variations shown in Figs. 8–10 arise from the prescribed perturbations in CWC and IWC, rather than from altitude differences alone. In other words, the observed reflectance differences mainly result from the distinct radiative contributions of clouds at different altitudes and with different microphysical properties (**Lines 297~303 in the revised manuscript**).

7. The oscillations in Figures 9c, 9d, 9f, 10a and 10b seems problematic. Especially, the peak in Figure 9f is not natural, and should be carefully checked.

**Response:** We thank the reviewer for the valuable comments. In the original version, the sensitivity experiments used a single CWC/IWC profile for each cloud type. Considering that different profiles within the same category could lead to different reflectance responses, we have revised the method in the updated manuscript by using the mean CWC and IWC profiles for each cloud category. This approach provides more representative and stable input conditions. The sensitivity experiments were repeated with these averaged profiles, resulting in smoother and more physically realistic outcomes, without the previously observed unnatural peaks and oscillations (**Lines 315~343 in the revised manuscript).**

To facilitate the reviewer's evaluation, the following provides a detailed description of the sensitivity experiment design, including the different vertical structures of double-layer and three-layer clouds, as well as the corresponding CWC/IWC profiles. For clarity, only the results of the sensitivity experiments are shown in the revised manuscript, while the detailed input profiles are summarized here for reference. The cloud structures were categorized into three types: (1) low-level water and high-level ice, (2) low-level water and mid-level ice, and (3) mid-level water and high-level ice. The CWC and IWC profiles used in ARMS are shown in Fig. S2(double layers) and S3(three layers).

**Lines 315~332:**

For double-layer clouds, the reflectance response is determined by the combination of liquid and ice clouds. In the "mid-level water cloud–high-level ice cloud" configuration, channel 5 reflectance remains nearly unchanged under different perturbations, whereas channel 2 exhibits significant variations. For CER < 5 μm, increasing mid-level CWC or high-level IWC yields almost identical results; for CER > 5 μm, increasing high-level IWC produces slightly higher channel 2 reflectance than increasing mid-level CWC (Fig. 9a, d), likely due to multiple scattering in the upper ice cloud enhancing upward radiation while partially diminishing the mid-level water cloud contribution. In the "low-level water cloud–high-level ice cloud" configuration (Fig. 9c and f), the overall reflectance response is similar to that of the mid-level water cloud– high-level ice cloud case, indicating that the upper ice cloud dominates the system's radiative properties. This mechanistically supports the observational findings of Kiran et al. (2015), which reported that despite the presence of liquid water clouds at lower levels, the net radiative forcing at the top of the atmosphere remains nearly balanced, primarily due to the simultaneous shortwave cooling and longwave heating effects of the upper ice cloud. By contrast, in the "low-level water cloud–mid-level ice cloud" configuration, the reflectance response to increases in CWC/IWC at different levels is markedly different from the previous two cases. Specifically, increasing low-level CWC causes channel 2 reflectance to vary nonlinearly with CER, first decreasing and

then increasing (Fig. 9b), whereas increasing mid-level IWC results in a linear decrease of reflectance in both channels 2 and 5, with channel 5 even decreasing by approximately 0.12 when CER > 30 μm (Fig. 9e).

[Figure]

Fig. S2. Sensitivity experiments of cloud vertical structures based on ERA5 CWC and IWC profiles. EXP1 (mid-level water + high-level ice): (a) original profile, (b) increased mid-level CWC, (c) increased high-level IWC. EXP2 (low-level water + mid-level ice): (d) original profile, (e) increased low-level CWC, (f) increased mid-level IWC. EXP3 (low-level water + high-level ice): (g) original profile, (h) increased low-level CWC, (i) increased high-level IWC.

[Figure]

**Figure. 9.** Reflectance–CER relationships for six double-layer cloud sensitivity experiments: (a, d) mid-level water + high-level ice cloud, with (a) increased mid-level CWC and (d) increased high-level IWC; (b, e) low-level water + mid-level ice cloud, with (b) increased low-level CWC and (e) increased mid-level IWC; (c, f) low-level water + high-level ice cloud, with (c) increased low-level

CWC and (f) increased high-level IWC. Each panel illustrates the effect of adjusting cloud water content (CWC) or ice water content (IWC) on the reflectance–CER relationship for the corresponding cloud vertical structure.

**Lines 333~343:**

For the three-layer cloud structure consisting of high-level ice cloud over mid-level water cloud over low-level water cloud, increasing either low- or mid-level CWC enhances Channel 2 reflectance with increasing CER, with the increase being more pronounced for mid-level CWC(Fig.10a-b). The response of Channel 5 reflectance exhibits a similar trend but with a smaller magnitude, indicating that the mid-level water cloud contributes more significantly to reflectance in both channels. When high-level IWC is increased, Channel 2 reflectance decreases approximately linearly with CER, while Channel 5 also declines (Fig. 10c). For CER >14 µm, Channel 5 reflectance drops from ~0.4 to 0.25, highlighting the strong radiative shielding effect of the high-level ice cloud on underlying water clouds. These findings align with Li et al. (2011), who reported that multilayer clouds generally have weaker shortwave reflectance than single-layer clouds due to partial transmission of radiation through high cloud tops to lower layers or the surface.

[Figure]

Fig. S3. Sensitivity experiments of three-layer cloud vertical structures based on ERA5 CWC and IWC profiles. (a) original profile; (b) increased low-level CWC (Exp 10) ; (c) increased mid-level CWC(Exp 9); (d) increased high-level IWC.

[Figure]

**Figure10.** Similar to Fig. 9, but for multi-layer clouds (a) Exp 9 and (b) Exp 10.

References:

[1] Ravi Kiran, V., Rajeevan, M., Gadhavi, H. et al. Role of vertical structure of cloud microphysical properties on cloud radiative forcing over the Asian monsoon region. Clim Dyn, 45, 3331–3345, https://doi.org/10.1007/s00382-015-2542-0, 2015.

[2] Li, J., Yi, Y. H., Minnis, P., Huang, J., Yan, H., Ma, Y., Wang, W., and Ayers, J.: Radiative effect differences between multi-layered and single-layer clouds derived from CERES, CALIPSO, and CloudSat data. J. Quant. Spectrosc. Radiat. Transf. 112(2), 361-375, https://doi.org/10.1016/j.jqsrt.2010.10.006, 2011.

---

## Author Comment (AC3)

**Response to the Reviewer2**

This study presents a bispectral retrieval algorithm for determining cloud optical thickness (COT) and cloud effective radius (CER) using data from the Advanced Geostationary Radiation Imager (AGRI) on the FengYun-4A satellite. The algorithm was validated against coincident MODIS cloud products, demonstrating its reliability. Furthermore, ten idealized multi-layer cloud scenarios were employed to investigate the sensitivity of visible and shortwave-infrared (SWIR) reflectance to vertical cloud structure. This study is interesting. I think this paper is publishable after several major revisions before I could recommend it with enthusiasm.

**Response:** We sincerely thank the reviewer for their constructive comments and valuable suggestions, which have helped us improve the clarity, coherence, and scientific rigor of our manuscript. We have carefully revised the manuscript according to the comments, and detailed point-by-point responses are provided below. All modifications are highlighted in blue in the revised manuscript.

Major comments:

1. This study comprises two primary components: i) the development of a CER and COT retrieval algorithm utilizing AGRI data, and ii) an assessment of the sensitivity of simulated visible/SWIR reflectance to vertical cloud configurations through ten idealized multi-layer cloud scenarios. That said, the linkage between these sections currently lacks immediacy. I suggest that improving the second part to detail how vertical cloud structures influence real-world retrieval outcomes would greatly enhance the paper's coherence and narrative progression.

**Response:** Response: We sincerely thank the reviewer for this critical insight regarding the linkage between the two primary components of our study. We fully agree that strengthening this connection is essential for the paper's coherence. Since the retrieval results serve as the basis for the sensitivity experiments, we have streamlined the retrieval sections in the revised manuscript by merging Sections 3.1 and 3.2 (**Lines 214–246).** The case study serves both to validate the retrieval model and, more importantly, to provide a basis for the subsequent sensitivity experiments. This connection is explicitly clarified in the revised manuscript (**Lines 230–231, 245–246**).

To further clarify this linkage, we have added a pivotal analysis in the revised manuscript (**Lines 385–411**) and present the key findings here. This new analysis compares the COT–CER relationships from multi-layer cloud simulations against those derived under the single-layer assumption, thereby directly detailing how vertical structures influence retrieval outcomes.

**The core results are as follows (and summarized in the new Fig. 12):**

The difference between single-layer retrievals and multi-layer simulations strongly depends on CER and vertical configuration ( $\triangle$ COT_retrieval, hereafter abbreviated as ΔCOT_R). Overall, when CER < 10 μm, ΔCOT_R changes from negative to positive, indicating that the single-layer assumption systematically underestimates the true COT under small droplet conditions. As CER increases beyond 14 μm, ΔCOT_R gradually becomes positive, with single-layer retrievals exceeding two-layer simulations by approximately 20 units on average. This primarily results from the single-layer assumption's inability to capture the shielding effect of overlying ice clouds on underlying water clouds, as well as the differential contribution of particles at different vertical levels to reflectance in the visible and shortwave infrared channels.

For the "mid-level water cloud and high-level ice cloud" structure, ΔCOT_R remains negative for CER < 22 μm before turning positive (Fig. 12a). For the "low-level water cloud and high-level ice cloud" structure, positive $\triangle$ COT_R appear only when CER > 45 μm (Fig. 12b). Increasing the IWC of the high-level ice cloud maintains negative ΔCOT_R at small CER, with the positive transition also at CER > 45 μm. In the "low-level water and mid-level ice" scenario, ΔCOT_R is near zero for CER < 5 μm, increases gradually with CER, and plateaus beyond 30 μm. The critical CER values where ΔCOT_R changes sign depend on the perturbed layer: ~14 μm for mid-level IWC and ~30 μm for low-level CWC (Fig. 12c), consistent with reflectance sensitivity results.

For the three-layer cloud case, increasing mid-level CWC results in single-layer retrievals being consistently smaller than the simulations when CER < 50 μm, highlighting the limitations of the single-layer assumption under complex vertical

structures (Fig. 12d). Together with the preceding COT–CER analyses, these results quantitatively demonstrate that neglecting vertical heterogeneity introduces significant biases in single-layer retrievals, with both the magnitude and sign of ΔCOT_R strongly dependent on CER and the vertical distribution, thickness, and microphysical properties of water and ice layers. Importantly, the trends observed in ΔCOT_R are consistent with the reflectance sensitivity experiments, confirming the direct impact of vertical cloud structure on operational COT retrievals.

[Figure]

**Figure 12.** Differences in cloud optical thickness (COT) between multilayer cloud vertical structures and the single-layer assumption as a function of CER. Blue: difference between COT retrieved under the single-layer assumption and COT simulated for double-layer clouds; Red and pink: difference after adding CWC to the mid-level water cloud; Green: difference after adding IWC to the high-level ice cloud.

2. In agreement with Reviewer #1, although Figure 4 demonstrates excellent consistency between AGRI-derived results and MODIS cloud products, the findings presented in Figures 5 and 6 reveal persistent limitations within the current retrieval algorithm. It is advised that the authors conduct a thorough reassessment of their results, ensuring that the representation in Figure 4 provides an unbiased depiction of the algorithm's capabilities.

**Response:** We sincerely appreciate the reviewer's comment. To further clarify this point, we have plotted the PDFs of COT and CER for MODIS (1 km) and FY-4A/AGRI (4 km) in **Fig. S1**. The results indicate that the overall trends of the two datasets are generally consistent, suggesting that the discrepancies observed in Figs. 4–6 mainly stem from differences in data processing and sensor resolution.

Since the pixel positions and spatial resolution of FY-4A/AGRI differ from those of MODIS, direct pixel-by-pixel comparisons are not possible. Fig. 4 shows a scatterplot based on FY4A/AGRI retrievals and MODIS data resampled to a 4 km grid, providing the overall pixel-wise correlation. Fig. 5 shows the PDFs of COT and CER aggregated on the same 4 km grid, illustrating the pixel-scale statistical characteristics. Fig. 6 displays the spatial distributions of COT and CER in the same region using the original-resolution data, showing consistent spatial patterns while highlighting differences caused by sensor resolution and retrieval methods (**Lines 214~220, Lines 221~225, Lines 230~237, in the revised manuscript**).

Previous studies have shown that cross-resolution data matching may introduce a "partial-filling effect." For example, when higher-resolution visible pixels (~2 km) are matched to coarser radar pixels (~5 km), clear-sky areas may be included, leading to shifts in the PDFs (Chen and Fu, 2017; Chen et al., 2020). Overall, the observed differences are mainly attributable to: (1) spatial resolution differences (MODIS 1 km vs. AGRI 4 km); (2) horizontal inhomogeneity of clouds within AGRI pixels; and (3) visible channel degradation and SWIR fluctuations (Sun et al., 2025). In addition, in the region of 106–107°E and 32–35°N (corresponding to the Dabie and Wuling Mountains), FY-4A/AGRI and MODIS retrievals exhibit noticeable differences (Fig.6), which may be related to the influence of high-elevation terrain on satellite observations. The current retrieval algorithm was primarily tuned for lowland surface types and does not explicitly account for mountainous characteristics. Despite these local discrepancies, the overall distributions of COT and CER remain consistent across the overlapping regions, thereby confirming the robustness and reliability of the retrieval method and providing strong support for the sensitivity experiments in Section 4(**Lines 238~246 in revised manuscript**).

[Figure]

Fig. S1. Probability density function (PDF) of the FY4A/AGRI retrieval results and the MODIS cloud products in the region. The red and blue solid line shows the FY4A/AGRI(4km) results and the MODIS(1km) results, respectively.

**References:**

[1] Fu, Y.: Cloud parameters retrieved by the bispectral reflectance algorithm and associated applications, J. Meteorol. Res., 28, 965982, https://doi.org/10.1007/s13351-014-3292-3, 2014.

[2] Ackerman, S. A., Holz, R. E., Frey, R., Eloranta, E. W., Maddux, B. C., and Mcgill, M.: Cloud detection with MODIS. Part II: Validation, J. Atmos. Ocean. Tech., 25, 1073–1086, https://doi.org/10.1175/2007JTECHA1053.1, 2008.

[3] Chen, Y., Chen, G., Cui, C., Zhang, A., Wan, R., Zhou, S., Wang, D., and Fu, Y.: Retrieval of the vertical evolution of the cloud effective radius from the Chinese FY-4(Feng Yun 4) next-generation geostationary satellites. Atmos. Chem. Phys. 20,1131-1145, https://doi.org/10.5194/acp-20-1131-2020, 2020.

Minor comments:

1. Line 132: As COT and CER were previously defined, repeating their definitions here is unnecessary.

**Response:** We appreciate the reviewer's careful reading. The repeated definitions of COT and CER at Line 132 have been removed and the sentence has been revised at **Line 150** to: "aiming to investigate the impacts of cloud layering on reflectance, COT, and CER."

2. Figure 1: Is the logical relationship depicted between "cloud detection" and "cloud" accurate?

**Response:** We thank the reviewer for pointing this out. The label "cloud" in Fig. 1 has been revised to "Cloud pixel" in the updated manuscript to accurately reflect that the detection is performed at the pixel level.

[Figure]

**Figure 1.** Framework of the COT and CER retrieval algorithm for FY4A_AGRI

3. Section 3.1: How are the 4-km AGRI observations/retrievals spatially matched with the 1-km MODIS cloud products?

**Response:** We appreciate the reviewer's question regarding the spatial matching of AGRI and MODIS data. To enable a fair pixel-by-pixel comparison, the 1-km MODIS retrievals were resampled to the 4-km AGRI grid using nearest-neighbor interpolation (**Lines 215~216 in the revised manuscript**).

4. Line 349: "10b" should be corrected to "11b".

**Response:** We thank the reviewer for pointing out this typo. It has been corrected from "10b" to "11b" in the revised manuscript (**Line 372**).

5. Figure 12: Lacks clarity in showing how visible/SWIR reflectance responds to vertical cloud structure.

**Response:** We thank the reviewer for pointing out the lack of clarity in Fig. 13(note that the original Figure 12 has become Figure 13 in the revised manuscript due to the addition of a new figure). In the revised manuscript, Fig. 13 has been improved to better

illustrate the response of visible (channel 2) and SWIR (channel 5) reflectance to cloud vertical structures. In addition, the related conclusions have been updated to better capture the implications of the revised conceptual diagram (**Lines 434~469 in the revised manuscript**).

Conceptual illustration of the response of visible (channel 2) and SWIR (channel 5) reflectance to vertical cloud structure. For single-layer clouds, low-level water clouds yield the strongest enhancement in channel 2 reflectance, while mid-level water clouds show a weaker effect and high-level ice clouds mainly increase channel 2 reflectance with little impact on channel 5. For double-layer clouds, the combination of low-level water clouds and mid-level ice clouds shows contrasting effects: increasing low-level water cloud enhances reflectance in both channels, whereas adding mid-level ice clouds reduces reflectance with increasing CER, especially beyond 30 µm. For triple-layer clouds, increasing low- or mid-level water clouds enhances reflectance, with mid-level contributions being more pronounced. In contrast, increasing high-level ice clouds leads to overall reductions in reflectance, particularly for CER > 30 µm in channel 2 and CER > 14 µm in channel 5, with the latter decreasing by up to 0.15.

[Figure]

**Figure. 13**. Conceptual diagram illustrating the radiative characteristics and retrieval implications of COT and CER under different vertical cloud structures. The thickness of arrows represents the relative magnitude of reflectance, while dashed lines indicate negative reflectance gradients with increasing CER. The numerical ranges in the figure denote the changes in reflectance when CWC/IWC increases from lower to high cloud layers.